# Resilience and Sustainable Urban Tourism: Understanding Local Communities' Perceptions after a Crisis

Ana Brochado [1,*], Paula Rodrigues [2], Ana Sousa [2], Ana Pinto Borges [3], Mónica Veloso [4] and Mónica Gómez-Suárez [4]

1 DINÂMIA'CET—Centro de Estudos sobre a Mudança Socioeconómica e o Território and Instituto Universitário de Lisboa (ISCTE-IUL), 1649-026 Lisboa, Portugal
2 COMEGI—Research Centre in Organisations, Markets and Industrial Management, Universidade Lusíada, 4100-346 Porto, Portugal; pcristinalopesrodrigues@gmail.com (P.R.); ferreira.antunes.ana@gmail.com (A.S.)
3 COMEGI—Research Centre in Organisations, Markets and Industrial Management, CICET-FCVC—Research Centre in Business Sciences and Tourism, ISAG-European Business School, 4100-442 Porto, Portugal; anaborges@isag.pt
4 Universidad Autónoma de Madrid, 28049 Madrid, Spain; monica.veloso@uam.es (M.V.); monica.gomez@uam.es (M.G.-S.)
* Correspondence: ana.brochado@iscte-iul.pt

**Abstract:** This study sought to examine the coronavirus COVID-19 pandemic's impacts on local communities whose residents are directly or indirectly affected by city tourism. Qualitative research was conducted via in-depth interviews and Leximancer software analysis to explore locals' perceptions in two highly tourism-dependent southern European cities. While the crisis has had predominantly negative impacts on tourism, the pandemic's positive effects could contribute to cities' greater resilience and more sustainable tourism models. The results highlight the variables that residents perceive as having the most influence on city tourism, as well as providing insights into locals' expectations for the future.

**Keywords:** local community; resilience; sustainable tourism; city; qualitative analysis; Portugal; Spain





## 1. Introduction

The coronavirus disease-19 (COVID-19) pandemic's rapid spread has made many of the last decade's most prevalent tourism strategies irrelevant (e.g., mass tourism and local accommodation proliferation). Locals' multiple complaints about overtourism have turned into a loud silence as tourist crowds have disappeared [1]. Examining the pandemic's effects on the tourism sector is quite important in order to learn from this crisis, so scholars have begun conducting research on this topic in different fields, including economics, social issues, and the environment [2–4].

Many studies have highlighted the pandemic's negative impacts on tourism. Local communities have suffered significant setbacks, including, among others, illness and death, income loss, job reductions, and decreased foreign exchange earnings. However, the pandemic may have also generated more constructive attitudes towards climate change, the circular economy, carbon neutrality, and environmentally friendly and resource-neutral strategies [5,6]. COVID-19 has thus clearly had massive negative consequences for many tourism stakeholders such as local communities, governments, and businesses, yet the crisis has given this industry an opportunity to pause and consider what these communities would like a post-COVID-19 tourism landscape to incorporate [7].

The term 'sustainable development' is widely used in almost every sector, although this goal has been criticised for its ambiguous definition and the assumptions on which it is based [8]. Sustainable development requires a holistic approach based on the four pillars of social, economic, ecological, and cultural strategies [9]. Given sustainability's emphasis on

conservation and impact mitigation, this word is being increasingly replaced by the term 'resilience' in tourism and other fields [10]. This concept involves adapting to change and understanding that tourism must be seen as a part of larger systems [11].

Lee et al. [12] reported that interest in social and community resilience as an alternative development model has been growing since the mid-2000s. Some scholars argue that sustainability and resilience are distinct conceptual paradigms [13], while others assert that these terms are quite similar and see resilience as an indicator of sustainability [14]. Still, other researchers consider sustainability to be the ultimate social goal and resilience to be the means by which sustainability can be achieved [15].

The present study adopted the latter approach. According to Sutcliffe and Vogus [16] (p. 95), resilience is 'the maintenance of positive adjustment under challenging conditions'. Coutu [17] (p. 50), in turn, maintains that resilience is based on a 'propensity to make meaning of terrible times' and that 'people build bridges from present-day hardships to a fuller, better constructed future'. Thus, this perspective attributes significant value to exploring how tourism can derive positive outcomes and/or meaning from the COVID-19 pandemic. In addition, local communities' resilience can provide opportunities to rebuild tourism using a triple-bottom-line strategy, which could ultimately produce stronger, more sustainable economies.

In tourism, resilience has been examined from the perspectives of community tourism planning [1,18], tourism in protected areas [19,20], employment [21], environmental governance [22], business sustainability [18], and integrated destination and disaster management [10]. Some studies have focused on local communities' resilience in tourism, but the COVID-19 crisis's effect on the perceptions of urban communities heavily dependent on tourism has not yet been specifically studied [13,23–26]. Previous research has assessed the impacts of municipalities' financial interventions to help residents cope with COVID-19 [27] and has explored various types of economic aid in different cities [28].

The current study sought to understand resilience from a different perspective as it concentrated on residents' perceptions of the pandemic's impacts on their city's tourism.

This research thus addressed a gap in the literature on the COVID-19 crisis's effect on local communities' resilience. More specifically, this study examined the crisis's perceived impacts on communities and residents directly or indirectly affected by city tourism. A flexible approach was adopted to analyse the pandemic's effects in greater detail. Qualitative methods were used to explore locals' perceptions in two major cities in southern European countries that heavily rely on tourism. This qualitative approach facilitated a comprehensive examination of residents' attitudes towards and perceptions of the most crucial initiatives in terms of building resilience in their cities.

Two research questions were addressed:

1. What concepts or variables do residents perceive as being at the core of the COVID-19 pandemic's main impacts on city tourism?
2. What do residents expect their city's tourism will look like in the future?

## 2. Literature Review

### 2.1. Sustainable Tourism Development and COVID-19 Pandemic's Influence

The COVID-19 pandemic has had a dramatic impact on tourism [29], with the travel and tourism sector's contribution (i.e., direct, indirect and induced impacts) to the global gross domestic product (GDP) shrinking by USD 766 billion in 2020 [30]. Given these adverse effects, various possible scenarios have been proposed. The first is that destinations affected by the crisis may, in the future, have difficulty attracting tourists, especially individuals who are risk-sensitive and fear being infected, because of these countries' damaged image. At least as long as risk exists of new outbreaks or waves, the number of COVID-19 patients in these destinations will affect intentions to travel to and within those nations [24,26,31]. The second scenario is that these destinations could benefit from tourists' charitable behaviour because they may choose to visit in part to provide economic support to residents [29].

In either case, local communities must prioritise building greater resilience in COVID-19's wake and aim for more sustainable tourism development in the future. As mentioned previously, sustainable development requires a holistic approach based on social, economic, ecological, and cultural strategies [9].

Residents' support is crucial because efforts to promote tourism despite the latest pandemic's effects and against locals' best interests could generate increased hostility towards tourists. Residents' negative attitudes can have a significant impact on tourists' perceptions by making them feel uncomfortable and unwelcome in holiday destinations [32].

According to previous research on tourism crises [33], community resilience positively affects locals' crisis response and community participation. Community participation might be viewed as a democratic process due to the development of local policies that foresee sustainability and resilience planning [34].

Engaging host communities in the tourism industry's recovery after a pandemic such as the present one would ensure that initiatives will promote residents' interests and could mitigate the crisis's devastating consequences [35].

An alternate view of the pandemic, in contrast to the current perspective worldwide, could be that this crisis is a moment of transition, e.g., [36]. That is, the pandemic may act as a catalyst for the implementation of regenerative practices and have a positive impact on the travel and tourism industries [37]. Tourism's benefits and disbenefits have been extensively discussed in the literature [14,38,39].

The more recent challenges presented by the COVID-19 pandemic have given new impetus to the debate about tourism and its affiliated sectors [4,29,40]. On the one hand, tourism supporters support these sectors' recovery rather than reforms and highlight their benefits, such as the valuable jobs and income they generate. On the other hand, advocates of limiting tourism have voiced their concerns about tourism's effects on the environment and ecosystems [41], human rights, inequality [42], and workers' [43] and local communities' rights [44].

Bianchi and de Man [38] specifically examined the role of the United Nations' World Tourism Organisation and Sustainable Development Goals, which focus on sustained and inclusive growth. The cited authors suggest that when the balance is lost between sustainability and long-term growth entailing tourism's expansion, this imbalance can produce significant negative outcomes, including more inequality and damaged natural habitats.

The COVID-19 pandemic presented new challenges that have highlighted the importance of reevaluating tourism with greater sustainability and resilience in mind. Experts have become more motivated to analyse health crises' effects on the economic, sociocultural, and environmental aspects of sustainable tourism development. In this context, resilience plays a critical role by reinforcing habitability principles and enhancing physical well-being and social welfare [45]. This perspective reinforces the need to go beyond sustainable economic and regional growth and embrace an approach that includes balanced ecosystems.

Ramirez Lopez and Grijalba Castro [46] suggest that public decision-makers lack adequate assessment tools to analyse their region's resilience, urban processes, and evolution over time [47]. Ramirez Lopez and Grijalba Castro [46] (p. 20) argue that:

> Sustainability and resilience are two concepts that will allow …strategic functionality in urban planning, because whereas the first one prioritises results, the second one analyses processes, demonstrating that a partnership between these two concepts will allow for a widening of …[planners'] focus to anticipate anthropocentric and natural uncertainties.

Analyses of the COVID-19 pandemic's impacts on sustainable tourism development must consider the relationships between different dimensions and specific topics. For example, various issues emerged within the social dimension. Social distancing measures and increased stress significantly affected people's well-being. Necessary responses included special measures such as emergency tourism packages, tourism product and service



diversification, and human capital development [48]. Higgins-Desbiolles [49] suggests that the COVID-19 crisis can thus be seen as a potential catalyst for essential transformations promoting more ethical, responsible and sustainable tourism and putting local communities centre stage by focusing on their needs and best interests.

The pandemic's economic dimension included its negative impact on tourism due to the temporary closure of most of the world's destinations. This meant that all the companies, whether large or small, that depend directly or indirectly on related sectors (e.g., air transport, cruises, hospitality, travel agencies, and leisure and cultural activities) ceased their operations [50]. The crisis's effect on the accommodation sector made guest numbers decline by 50% or more in all countries [3].

One of the most-likely long-term consequences is the bolstering of proximity tourism [51], which is understood as tourism and travel near one's home [52,53]. In addition, sustainable tourism development in the future will depend on tourists' behaviour, destination choice, and tourism plans. The participation of other stakeholders (e.g., local communities, tour operators, accommodation owners, and transportation providers) will also be crucial [54].

Finally, the pandemic's environmental dimension and its impacts on tourism include that, in contrast to the negative economic and social effects, measures such as travel restrictions and the related reduction in economic activities have produced a short-term improvement in global air quality. Air and water pollution has also been reduced, which has had health benefits for the public [55]. The restrictions and controls placed on human mobility have demonstrated that tourist flows can be regulated to satisfy sustainability standards [50]. Therefore, the COVID-19 crisis has offered valuable lessons to policymakers, academics, and the tourism and hospitality industries' players concerning global change's effects.

To achieve genuine sustainability, tourism models must be conceptualised within the context of long-term sustainable development. Higgins-Desbiolles [49] (p. 565) asserts that 'we [should] put tourism in its place, at the service of local communities and societies. Tourism is not an end in itself; thus, sustaining tourism is not the ultimate goal.' That is, myopic perspectives must be avoided to prevent social and environmental injustices. Sustainability is of utmost importance as it fosters destinations' competitiveness and improves their socioeconomic conditions [56]. Another important outcome is related to sustainability through decreased tourism, with a focus on local communities' needs and best interests [44]. This approach responds to Ateljevic's [37] call for transformative travel and tourism, which involves looking for a balance between the environment, societies and economies, and living more authentically in greater harmony with nature and humanity.

### 2.2. COVID-19 Pandemic's Impact on Local Communities

Prior research has demonstrated that local residents' support for tourism development depends on their positive and negative perceptions of tourism's effects on their community [57–59]. This support influences destinations' sustainable development and determines locals' behaviours and attitudes towards tourism activities [60]. According to social exchange theory, host communities' tendency to back tourism development projects is based on residents' perceptions of these initiatives' costs and benefits [61]. If they see tourism's benefits as more significant than its costs, locals will be more willing to support tourism's expansion [62].

In addition, some scholars have assessed host communities' perceptions of sustainable tourism based on its perceived economic, sociocultural, and environmental impacts [63–65]. Other researchers have included life satisfaction and well-being as measures of residents' perceptions [66]. Cottrell et al. [67], in turn, found that all of sustainable tourism's dimensions are predictors of resident satisfaction, but the economic dimension is the strongest determinant of positive attitudes as opposed to the sociocultural and environmental dimensions.

In general, tourism's positive economic impacts are related to new employment opportunities, an improved standard of living, inward investment, and local businesses' profitability. Negative economic effects have been identified as increases in the cost of living and the industry's seasonality and creation of dependency [65,68,69]. Positive environmental impacts have been detected in the form of natural area protection, environmental awareness, and better environmental management, while negative effects are pollution, overcrowding, and facility and service saturation [65,70,71]. The positive sociocultural impacts identified comprise, for example, local cultures' promotion and preservation, greater diversity and tolerance, and improved quality of life. The negative consequences include, among others, increased crime and drug abuse, cultural conflicts between locals and tourists, and local residents feeling that their area has lost its authenticity [57,71].

Hitherto, academics have argued that local communities' perceptions of tourism have been influenced by specific factors, such as the level of tourism development [66], distance from specific tourist attractions, frequency of resident-tourist interactions [62], sociodemographic variables, and types of tourism [60,72]. The COVID-19 pandemic is also likely to have changed locals' views about tourism, so researchers need to identify how host communities' perceptions of tourism development's positive and negative impacts have been affected by the COVID-19 crisis. For instance, studies have shown that local communities have become less concerned about overtourism and that they believe that the benefits can outweigh the costs, which may make residents feel more inclined to support tourism development [73].

Restrictions have been imposed by governments to decrease the number of contagions, such as limiting international, regional, and local travel, putting communities on lockdown, requiring social distancing, and reducing bars and restaurants' capacity. These measures have had a strong negative impact on local communities, as many local businesses have been forced to close temporarily or even permanently [74].

Previous economic crises have also lessened residents' negative perceptions of tourism [26] and thus increased local support due to significantly lower estimations of tourism development's costs [58]. During economic crises, residents prioritise economic benefits over any perceived sociocultural and environmental costs. In other words, individuals prefer to sacrifice the environment and put sociocultural values to the side in order to receive economic benefits [75].

However, the current situation generated by the COVID-19 pandemic could lead to other critical subconscious changes in both tourists and locals' behaviours [28]. Tourists' altered behaviours may include a stronger focus on their home country and/or local tourism, an aversion to contexts characterised by massification and the avoidance of overcrowded destinations in favour of remote and/or rural places, among others. Changes in residents' behaviours are less easy to predict as locals must maintain a balance between financial considerations and tourism's possible negative impacts on their quality of life. In particular, new COVID-19 outbreaks will probably occur as a result of tourism [32].

Various authors have thus posited that residents may become less welcoming to tourists and less supportive of tourism development, even to the point of displaying xenophobia [13,29]. In tourism contexts, xenophobia has been defined by Shahabi Sorman Abadi et al. [75] as 'the amount of anxiety and discomfort that either residents associate with tourists or ... tourists perceive while traveling to and from destinations'. Historically, pandemic diseases have often been associated with ethnic outsiders, which has caused mainstream local groups to reject minorities and foreigners in order to avoid new outbreaks [76]. Researchers need to focus more strongly on residents' attitudes towards tourism development to discover ways to resolve the dilemma between welcoming tourism's benefits and fearing the appearance of new outbreaks, which arise upon tourists' arrival during pandemics such as the current crisis.

*2.3. Resilience-Based Approach to COVID-19*

Given the crisis's undoubted negative impacts on all levels, many experts agree that greater resilience is needed to take advantage of this situation's potential benefits, address pre-existing problems such as massification and climate change, and introduce appropriate transformative measures [77]. Community resilience has been studied in various contexts and from numerous perspectives [78]. This concept developed out of theoretical research on socioecological resilience, which characterises community resources as a social or socioecological system [62].

Resilience has been described as having the ability and knowledge to cope with natural disasters, social and political conflicts, and climate change and to manage sustainable ecosystems [13]. This concept has been integrated into efforts to maintain local communities' integrity through empowerment and sustainability [13,79]. Resilience has also been defined as social systems' capacity to absorb disruption and restructure while retaining most of their functions, structures and identity [13,79].

Strengthening community resilience involves adjusting communities' social lives and environments to ensure they can withstand multiple shocks [79]. Chen et al. [78] (p. 606) observed that a 'resilient community is better able to survive, adapt to, and occasionally transform itself in the face of unexpected and uncertain environment change'. Adger [80] further defined social or community resilience as local communities' ability to withstand external shocks to their social infrastructure.

According to Lee et al. [12] (p. 21):

[Resilience is] about adaptation, including building human resource capacities to change in efficient ways, creating learning institutions that can address changing circumstances while maintaining core values, understanding feedback loops in dynamic social and environmental systems and generally encouraging flexibility, creativity, and innovation in the culture of a community.

Thus, a new goal has emerged during the COVID-19 pandemic: to create sustainable and resilient local communities that embody enduring strength and vision [12].

## 3. Materials and Methods

*3.1. Research Context, Background, and Sampling Procedure*

Spain and Portugal have long experienced a high demand for their tourism offer, so this industry is an important contributor to their GDP. In 2019, tourism revenues made up 12.4% of the Spanish GDP, and related sectors provided 2.72 million jobs or 12.9% of total employment [81]. In Portugal, tourism is the largest foreign exchange earner, accounting for 52.3% of service exports and 19.7% of total exports in 2019. Tourism revenue in 2019 contributed 8.7% of the national GDP, and this industry provided 6.9% of all jobs [82].

This study concentrated on residents of Portugal and Spain's two largest cities. Madrid and Barcelona were chosen as the Spanish cities for this study as they are the two most visited and populous Spanish cities. Madrid has 5,743,402 inhabitants, and Barcelona has 6,779,888 [81]. Their rivalry is well-known [83]. Following the same reasoning, the Portuguese cities selected were Lisbon (1,083,050 inhabitants) and Porto (837,555 inhabitants) [84]. The tourism industry in these four cities has suffered great losses due to the COVID-19 pandemic.

As this crisis is quite recent and cities and individuals are still immersed in its daily realities, this research adopted a qualitative exploratory approach. Qualitative studies are useful for expanding knowledge about specific situations by understanding participants' experiences, expectations, and preferences [85]. The selected approach addressed Jarness's [86] call for research utilising qualitative methods to sift through cross-class interactions, which can be expected in various population segments. These methods enable analyses that produce a complex, holistic picture of participants incorporating detailed, in-depth descriptions of their viewpoints [87] based on a sequence of reinterpretations and literature-based corroborations.

The primary data collected for the current research were mostly information that could help explain residents' interrelated needs during the COVID-19 pandemic, which, while largely emotional, can also be social, cultural, and rational. The data analysed were gathered from a series of in-depth interviews. This technique was considered to be the most appropriate to meet the study's objectives; that is, to understand and explain residents' individual viewpoints on their community's interconnected tourism-related needs during the pandemic.

To avoid cross-contamination during the analytical procedures, each research team member took on a separate role in each phase. The research design and participant recruitment were carried out by two team members, the interviews and transcript preparation by two other researchers, and the analysis was conducted by a researcher who had not participated in the earlier phases. At the end of this process, the entire team came together to analyse the results and prepare this paper.

Scholars have previously recommended that qualitative studies be based on a minimum sample size of at least 12 participants to achieve data saturation [88,89]. The final sample size was thus determined by thematic saturation: the point at which additional data appeared to contribute nothing new to the findings, as shown by the interviewees' repetition of the same themes and comments [90]. Data generation was terminated at this point. The in-depth interviews were conducted in the two selected countries, namely, six in Spain and six in Portugal. This number per country provided enough data to compare the results for both countries.

Appendix A provides more details regarding the sample structure. More specifically, the interviews were set up to ensure various sociodemographic characteristics were adequately represented, so the sample incorporated different categories of residents. The interviewees were carefully selected to ensure a variety of opinions and to avoid bias. That is, the participants' profiles included different combinations of three sociodemographic variables (i.e., gender, age over or under 30, and education) that could affect the interviewees' perceptions of the phenomena under study. This process ensured that the sample was composed of varied population segments that would provide diverse responses reflecting dissimilar and/or similar points of view.

*3.2. Research Design*

A detailed discussion guide was developed that was discussed and agreed upon with the interviewers in order to help them conduct effective interviews. The questions were open-ended and designed to trigger a broad discussion of tourism's importance to local communities, as well as being mostly based on the literature review presented in Section 2. The questions were combined with projective exercises, a mixed method technique that provided the qualitative data needed to analyse the interviewees' unconscious and conscious motivations and the rational, objective reasoning underlying their statements.

First, basic sociodemographic data were elicited (i.e., detailed professional occupation and education descriptions). In the interviews' main part, the introductory section focused on general opinions about the participants' professional activity and tourism's role in their city's development. The second section consisted of open-ended questions regarding five main concepts (i.e., sustainability, smart cities, ill-being, well-being, and resilience). The third section was devoted to discerning how the pandemic has affected their activities, businesses, and, in particular, their city and country's levels of economic activity. The last section comprised questions to draw out their personal opinion about the future of tourism in their city.

The interviews lasted an average of one hour. The fieldwork was carried out from 10 November to 12 December 2020. The participants were informed of the research objectives and asked to take part in the study on a voluntary basis. The interviews were videotaped after the interviewees had given their permission. Complete transcripts were extracted from the video recordings.

The data analysis sought to contextualise or establish the meaning of the participants' statements and then to test the strength of the relationships between the concepts identified and their definitions. The analytical procedures also focused on grouping the concepts into higher-level constructs and constructing maps that could help researchers develop models in future studies.

The interview responses were organised according to the four main blocks of questions (see Appendix B). Sections 1 and 2 included introductory questions requiring short responses from the participants, which helped gain the interviewees' confidence and steer the subsequent discussion. Sections 3 and 4's questions were the core of the interview as they were designed to explore the main dimensions related to the two research questions. This paper specifically discusses the results obtained with two questions focused on the two predefined objectives (i.e., P11: In your opinion, how has the COVID-19 pandemic affected tourism? [Q11]; P12: Finally, what, in your opinion, will tourism be like in your city 10 years from now? [Q12]).

*3.3. Data Analysis*

The 12 interview transcripts were processed using content analysis based on a combination of Leximancer software functions and narrative analysis [91]. The automated procedures involved unsupervised quantitative content analysis of natural language electronic texts based on Bayesian statistical theory, nonlinear dynamics, and machine-learning algorithms [92]. More specifically, Leximancer facilitated an inductive identification of themes, with minimal manual intervention, based on a two-step approach: conceptual analysis (i.e., frequency of concepts) and relational analysis (i.e., co-occurrence between concepts). This software's lexical text analysis tools have also been shown to offer reliable results [93] consistent with grounded theory methodology [94,95].

This software also generated graphical representations of the concepts, each indicated by a node, and clustered these into themes indicated by larger shaded circles. Concepts that appear closer to each other on the map were found more frequently together in the interview transcripts. The number of themes present was determined using Leximancer's default option. Namely, the maximum theme size is equal to 33% of the concepts. Finally, following Brochado et al. [96] and Moshin et al. [91] lead, narrative analysis was performed to identify the source text segments that contained specific topics that referred to each concept [97].

## 4. Results

*4.1. Research Question One*

The first research question was as follows: What concepts or variables do local residents perceive as being at the core of the COVID-19 pandemic's main impacts on their city's tourism? The analysis identified 13 themes in the interview data: city (tourism), hotel, airline, unemployment, hospitality, atmosphere, social (issue), homeless(ness), planning, reinvent(ion), (saturation) aware(ness), local (tourism), and (solid) waste. These themes were classified according to two variables, the type of impact (i.e., economic, sociocultural and environmental) and the impact's sign (i.e., positive or negative), based on narrative analysis and the literature review's findings (see Figure 1). Some themes can be associated with both negative and positive effects.

The themes of city (tourism), hotel, airline, unemployment, and hospitality encapsulate the main perceived negative economic impacts. Atmosphere, social (issue), and homeless(ness) are all also negative themes but are related to sociocultural effects. Although most of the themes are perceived as negative, the interviewees identified some positive impacts, namely, planning, reinvention, support for local tourism, and less solid waste, which contributes to positive environmental change. Figure 1 above presents the map of the main themes and their concepts.

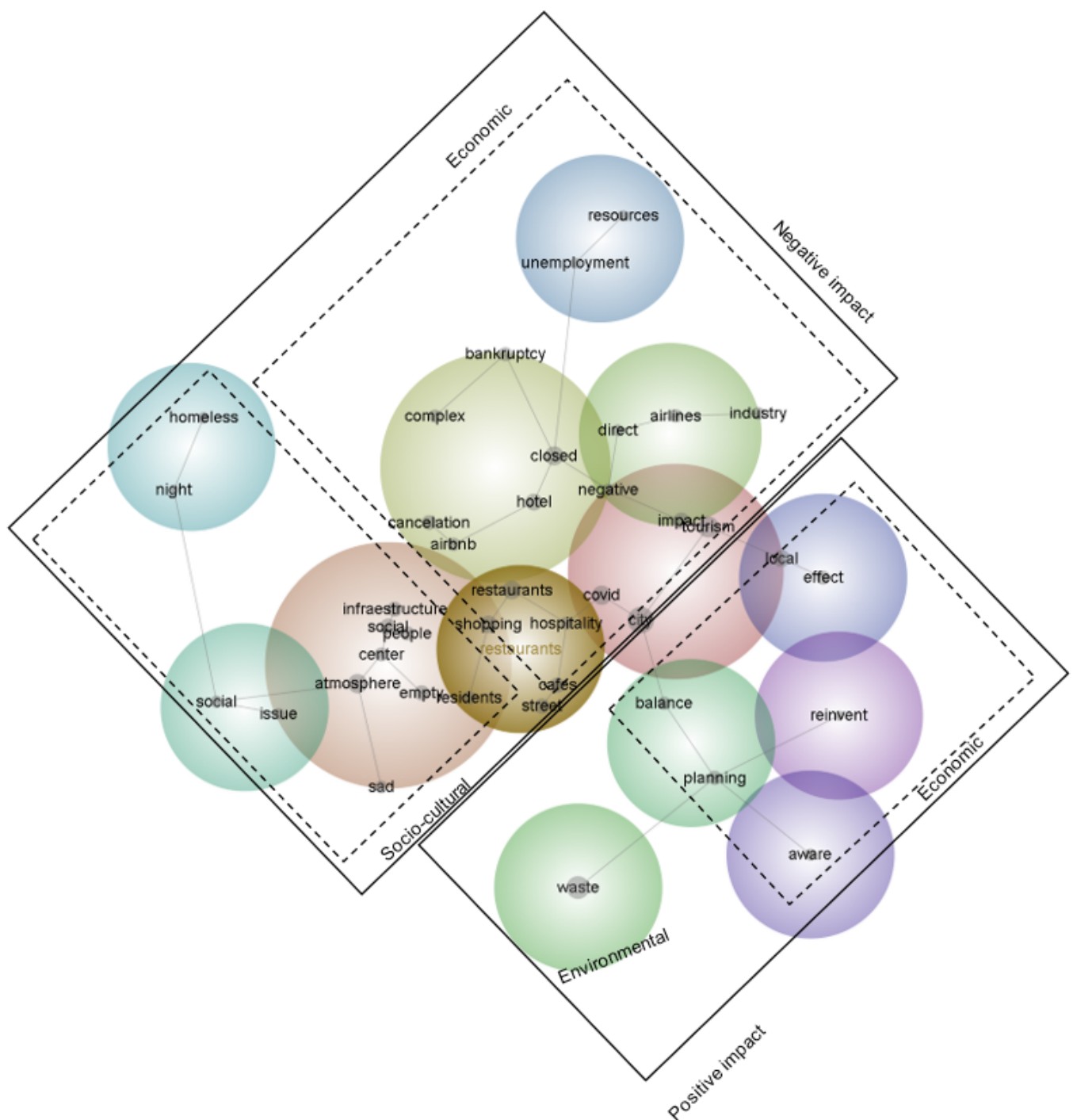

**Figure 1.** Concept map.

The first theme of city (tourism) includes the specific concepts of COVID, its impact, and its negative (effects). One participant agreed that '[t]he impact [of the pandemic] was negative' (Lisbon 2). Another interviewee observed that impacts, although negative, can vary by city:

Porto experienced fewer negative impacts from COVID on tourism than, for example, a city like Lisbon. . . . The queues of people . . . are the [main] indicator of tourism in each city. And there never failed to be . . . a considerable number of people visiting that particular part of the city [centre of Porto] (Porto 2).

A further participant noted, '[t]hat logically it [tourism] has been noticeably reduced, especially cross-border tourism as the borders were closed … with the levels of contagion that exist throughout the European Union' (Madrid 3).

The theme of hotel links the concepts of accommodation, cancelation, closed [hotel], bankruptcy, and complex [situation]. The interviewees reflected at length on reservation cancellations. One person said, 'even when prices were considerably reduced and reservations were made, later they were cancelled' (Porto 1). Another interviewee lamented, 'it's been cancellation after cancellation, loss after loss, expense after expense' (Porto 2).

Regarding accommodation, the crisis has evidently created different business opportunities. In particular, local property investors are trying to convert properties into student accommodations. A participant reported, 'we know that perhaps 50% of local accommodations are empty and [yet] people have been looking for long-term accommodations, rentals and long-term rentals' (Lisbon 1). Another response was, 'now we're going to discover whether students will want to live in these properties, but the houses are built for locals. I do not see this approach as being effective' (Porto 3).

The theme of airline includes the concepts of industry and direct (impact). A participant stated, 'there has been a direct impact on the entire sector, starting with airlines. … It seems to me that this is the most affected industry right now' (Madrid 1).

The theme of hospitality comprises the concepts of restaurant, cafe, street, shopping, and resident. This theme's narratives offer useful summaries of the COVID-19 crisis's effects on hospitality businesses. One individual observed, 'the issue of COVID-19 is fatal for …, for example, the hospitality industry' (Barcelona 2). A further interviewee shared, 'I see the empty streets. The streets are empty, the terraces are closed, the restaurants are closed, the pastry shops are closed' (Lisbon 1). Another participant (Lisbon 2) reported:

It's a shame to see the shops and restaurants empty. I see this as negative. If, on the one hand, for example, as I mentioned [before] …, the city was completely crowded with tourists, and that was bad, … now there are too few.

The theme of hospitality is closely connected to the theme of atmosphere. The interviewees shared their feelings about their cities' 'emotional environment' during the COVID-19 crisis, using keywords such as sad(ness), centre, empty (street), people, social (issue), and infrastructure. One participant felt that 'if you go to that shopping street and the restaurants are not open, it is soulless, that is, there is a lack of atmosphere, which was sad' (Barcelona 2). Another person (Porto 2) said:

I went, for example, to the downtown, historic area of Porto, and it was a truly scary image because, visually, my mind was not prepared, in the full light of the day on a weekend, to see those streets so completely deserted.

The theme of social (issue) covers the single concept issue; that is, problems that need to be resolved. An interviewee asserted that 'COVID has had this great negative impact from both an economic point of view and a social point of view' (Lisbon 2).

An unexpected theme that is extremely important from a social issue viewpoint is homeless(ness), which includes just one concept: night. One participant (Porto 2) noted that:

People are economically worse off, and there is a huge increase in homelessness. … [I] find it hard to see people on the street …, so now it makes me feel very bad going into Porto at night.

Regarding the pandemic's positive impacts, the theme of planning includes the single concept balance, which is linked to opportunities to make plans for the future that consider both tourists' and local residents' needs. An interviewee (Lisbon 3) said:

> [T]ourism planners think about the best way to move forward and correct what hasn't been so good in the past. … One story that we heard concerned a local resident who wanted to continue living in the same historic, traditional area and [who couldn't because of] tourists who wanted to experience living in an old Portuguese house.

Another participant asserted, 'we must find a balance between what there was and what there should be' (Barcelona 1).

The theme of (saturation) aware(ness) is closely related to the theme of planning, highlighting that city planners should be aware of saturation problems that existed before the COVID-19 crisis. One interviewee argued that city planners must 'recognise the saturation scenario we were experiencing. . . . The positive thing is that this awareness will help in the future when we return to a normal situation' (Barcelona 1).

The theme of reinvent(ion) is linked to the need to adapt new business models. A participant (Madrid 1) averred:

It is true that we have to reinvent ourselves . . . , for example, hotels that have been made into restaurants and events that are now being held online, . . . and take advantage by setting up different types of business.

Regarding the theme of local (tourism), one interviewee (Madrid 3) suggested:

[P]erhaps, in return, COVID-19 has had a positive effect on local tourism as, despite the restrictions, it could have been the best year for tourism in Spain because many areas have been practically empty, which is ideal for some forms of tourism.

The theme of (solid) waste is linked to environmental issues. One participant noted that, before the COVID-19 pandemic, 'a great burden was the solid waste generated [by tourism]' (Lisbon 3).

### 4.2. Research Question Two

The second research question was as follows: What do residents expect their city's tourism will look like in the future? The analysis revealed eight themes that address this question: tourism, change(d accommodation), hope, learn(ing), sustainability, culture, management, and balance (see Figure 2).

The theme of tourism comprises the concepts city, economy, (tourism project) developed, (technology-)based (city), environment, technology, and population. The narratives related to this theme focus on smart cities and smart tourism. One participant said, 'the city of the future that I imagine is a city that maintains a balance between technology and the environment . . . but with more development.' Another interviewee envisioned 'a greener city [that will be] bluer, without pollution, smart, conscious, responsible and attractive to the [local] population' (Madrid 2).

The theme of change(d accommodation) integrates the concepts of accommodation, building, housing, tourist, area, and important (features). The interviewees expect to see changes in their city's accommodation sector, with more opportunities for local residents in terms of new houses and lower prices in the city centre. One participant commented that 'buildings cannot all become local accommodation blocks. . . . City residents should continue to occupy their homes' (Porto 1). Another interviewee reported that 'young people have had to leave the cities' (Barcelona 3), although a further individual said, 'many accommodations have already been converted into long-term rentals' (Lisbon 2). Overall, most participants suggested that, in the future, 'local tourist accommodations may go back to housing' (Lisbon 2) and 'that property owners will need, in the future, to think carefully before . . . they change how spaces are used' (Lisbon 1).

The theme of hope covers the concepts recover, return, different (future), and everything. The interviewees hope that travellers' trust will return and increase the demand for tourism services. One participant predicted that '10 years from now, . . . I think tourism will return [to normal]' (Porto 1). Another person observed that 'people are afraid of flying. . . . We are going to lose this fear of flying and thus tourism is going to recover' (Barcelona 3).

The theme of learn(ing) encompasses the concepts learn(ing) and gain. An interviewee voiced the hope 'that we learn from everything that is happening and that the crisis leads to more controlled tourism' (Barcelona 1). A further participant said, 'the city will gain a lot [from the crisis]' (Lisbon 1).

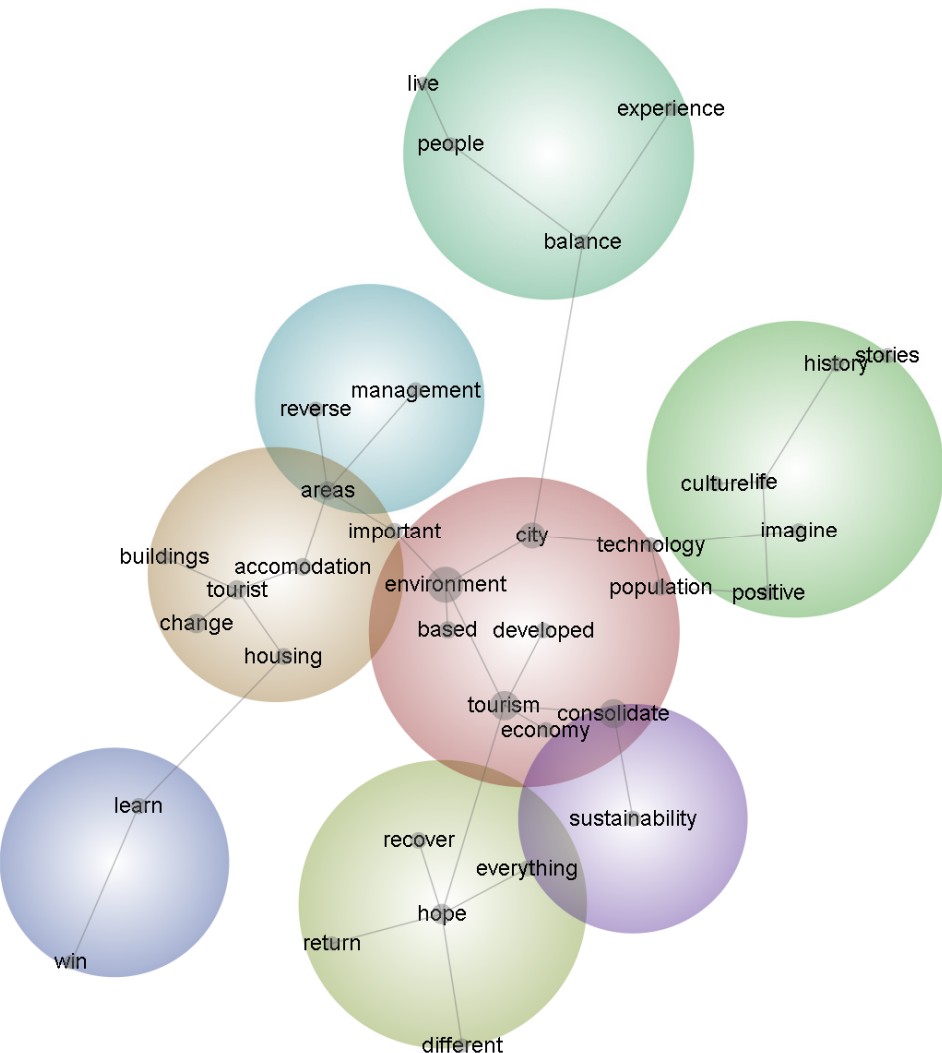

**Figure 2.** Tourism in the future.

The theme of sustainability links the concepts of hope, change, and consolidate (d tourism). One resident predicted that tourism 'will consolidate [and] will improve ..., gaining consistency and sustainability' (Porto 1), while another interviewee said tourism will 'enter a phase of maturity and sustainability' (Porto 2).

The theme of imagine (d city) encompasses the concepts of imagine (d city), positive (image), life, culture, history, and story. This theme is related to the cities' brand image and identity. One participant hoped that the city would 'not distort its identity and culture in the face of mass tourism' (Porto 2). Another individual argued that a 'city that is proud of its history ... should ensure that its tourism offer increases awareness of that history' (Lisbon 3).

The theme of management includes the concepts of management and reverse and has a close connection to accommodation issues and/or local accommodations. An interviewee suggested that 'perhaps ... some municipal [strategic] management can actually reverse this situation' (Lisbon 2).

The theme of balance is linked to (place to) live, people and experience. One participant shared that, 'when I go to see places, I want to observe and I want to experience what the place has to offer its inhabitants' (Lisbon 1). Another resident expected Porto to have the 'ability to maintain a high standard of quality in its [tourism] offer and achieve a balance between offer and demand' (Porto 2).

## 5. Discussion

This study sought to understand how the COVID-19 pandemic has affected local communities' perceptions of tourism. Data collected from 12 in-depth interviews in Portugal and Spain were subjected to content analysis to address two research questions.

With regard to the first question (i.e., What concepts or variables do residents perceive as being at the core of the COVID-19 pandemic's main impacts on their city's tourism?), the analysis identified 13 themes that were classified either as positive or negative effects. Similar to previous studies [3,25,98], the present results show that locals perceive the COVID-19 crisis's impacts on tourism as mainly negative. Adverse effects are also mentioned in connection with the travel and airline sectors. These sectors should thus be integrated into any new international monitoring and rapid response plans to reflect a better understanding of tourism's role during pandemics [3]. The current study also identified other previously unreported negative social effects, such as the sad atmosphere found in cities' deserted streets, increased poverty, more unemployment, and a greater number of homeless individuals.

However, the COVID-19 pandemic can also be seen as an opportunity to rethink tourism's growth trajectory, so this research identified some, although fewer, positive aspects. These effects include the need for more planning in the future and tourism's ability to reinvent itself, which could establish a better balance between tourists' and residents' needs. Another positive impact is greater awareness of how the movement of people around the world can contribute to pandemics' rapid spread, a new permanent reality that requires all tourism stakeholders to pay closer attention to health issues [99]. The reduction in pollution and waste is an additional positive effect of the crisis [100].

Regarding the second research question (i.e., What do residents expect their city's tourism will look like in the future?), the content analysis found eight themes. The locals' predictions of how tourism could develop in the future provide interesting insights. First, residents hope to benefit from the development of smart cities and smart tourism. Innovation in tourism was previously highlighted by Hall et al. [101] and Sharma et al. [13] because of technology's important contributions to creating greater flexibility in tourism services. New technologies offer varied benefits to consumers [102] and, in tourism contexts, serve utilitarian purposes valued by both business managers and visitors [103]. Technological innovations have changed traditional paradigms in the hospitality and tourism industry by facilitating authentic tourism experiences and greater competitiveness in tourism markets [104].

Second, the locals' narratives refer to the concepts of sustainable and equitable tourism [105,106] because the participants hope that the COVID-19 pandemic will lead to changes in local accommodation arrangements so that more people can return to live in city centres. Another interesting result was that residents expect travellers to regain their trust in tourism destinations and thus increase tourism demand [107].

The present results also focus on the need for a better balance in the future between residents' and tourists' needs [105], underlining the importance of, among other things, quality of life, the environment, health, and culture. Finding this balance could be a key lesson learned from the COVID-19 pandemic [13]. Locals thus want city planners to use a more community-centred framework that emphasises strengthening resilience through adjustments in communities' social life and environment, as previously recommended by McCartney et al. [79] and Sharma et al. [13].

Municipalities need to implement management strategies that create and develop a city brand image and identity-based on local communities' culture, history, and lifestyle. These neighbourhoods must be understood as epicentres of the transformation process triggered by the pandemic [13]. Paunovic and Deimel [33] further suggest that an intergenerational dialogue is required regarding perceived brand image attributes and development priorities to enhance residents' participation and their communities' sustainability and resilience.

## 6. Conclusions

The most recent research focusing on the COVID-19 pandemic's effects has examined municipalities' financial interventions to help locals cope with the crisis [27] and has explored various types of economic aid in different cities [28]. The present study contributes to this literature by providing a different perspective based on residents' perceptions of the pandemic's impacts on their city's tourism.

### 6.1. Theoretical Contributions

The interviews analysed offered new insights into the pandemic crisis's current and future impacts from the locals' perspective. The concerns expressed were transversal, involving the labour market (i.e., unemployment and job opportunities) and social issues (i.e., security, homelessness, and social instability). The pandemic has also had economic effects (i.e., closure of tourism companies, restaurants, accommodations, and complementary services) and affected transport systems, especially airlines.

The results additionally reveal that residents think that the COVID-19 crisis has generated new opportunities for tourism destinations. The common denominator in the interviewees' statements about positive impacts is that, from here on out, cities must adopt tourism models that follow smart city and tourism principles and meet locals' needs. Residents further emphasise sustainability's role in maintaining their quality of life.

### 6.2. Practical Implications

The COVID-19 pandemic's impacts on residents are significant, so interdisciplinary research on the ensuing issues should be conducted [108]. Studies need to focus on identifying tourists' and residents' perceptions of the crisis's effects in order to help policymakers develop measures that minimise the damage caused by the pandemic's negative consequences [109] and that reinforce COVID-19's positive impacts. The present investigation confirmed that these perceptions must be understood more fully to develop strategies that mitigate negative effects and, conversely, strengthen positive impacts. Destination managers should use this opportunity to pause and reset the tourism industry using a more sustainable, equitable model of tourism.

This study's findings thus have managerial implications, including that smart cities' tourism management strategies should be based on innovation, technology, sustainability, accessibility, and housing availability. If these factors are considered, destinations can improve their residents' quality of life and become competitive again in the global tourism market. More specifically, a balance needs to be found between locations as tourism destinations and as good places to live and experience. Locations that have succeeded in these two areas are already gaining recognition as smart tourism destinations.

The above results highlight the need to build a community-centred tourism framework by taking responsible steps towards re-establishing a tourism offer that benefits locals. The goal should be to avoid simply returning to pre-COVID-19 tourism practises and instead to develop a tourism model driven by not only demand but also an ethic of caring. This model must include social and environmental justice and ethnic reconciliation to support locally owned and managed tourism companies and minority residents [105].

In addition, the COVID-19 pandemic has underlined how locals need to be able to overcome the negative impacts on tourism-related businesses and develop greater resilience. The latter should be understood as communities' ability to reimagine and reinvent their role in this industry. Applying a resilience-based approach will help residents learn how to deal with similar crises yet to come.

### 6.3. Limitations and Avenues for Future Research

Regardless of these significant contributions, this research had some limitations. Yeh [102] suggests that because the pandemic took place during the present investigation's fieldwork, the findings cannot reveal the full scope of the crisis's repercussions. The four cities selected are all located in southern Europe, so the results may not be directly

translatable to other cities in the European Union. A deeper understanding is also needed of interviewees' perceptions of information and communication technologies' role in the development of sustainable tourism. Given the uncertainty surrounding the pandemic and its impacts on tourism, comparative studies should be carried out in other European Union cities to evaluate each city's results in view of the present findings.

Due to the nature of qualitative analysis, the results discussed in this paper cannot be considered statistically representative. Thus, future quantitative-based research based on surveys of different cities' residents combined with multi-country analysis could provide fresh information about the COVID-19 pandemic's impacts on community-centred tourism frameworks and resilience in other European cities.

**Author Contributions:** Conceptualization, A.B.; Paula Rodrigues, A.S., A.P.B., M.V. and M.G.-S.; Methodology, A.B., P.R., A.S., A.P.B., M.V. and M.G.-S.; Formal analysis, A.B., P.R., A.S., A.P.B., M.V. and M.G.-S.; Writing—original draft, A.B., P.R., A.S., A.P.B., M.V. and M.G.-S.; Writing—review & editing, A.B., P.R., A.S., A.P.B., M.V. and M.G.-S. All authors have read and agreed to the published version of the manuscript.

**Funding:** This work was supported by national funding of FCT—Fundação para a Ciência e a Tecnologia, I.P., in the project "UIDB/04005/2020".

**Institutional Review Board Statement:** Not applicable.

**Informed Consent Statement:** Informed consent was obtained from all subjects involved in the study.

**Conflicts of Interest:** The authors declare no conflict of interest.

## Appendix A

The interviewees were selected based on an assessment of their characteristics. This study relied on a heterogeneous purposive sample designed to reveal key terms [110], so the applicant selection criteria included two filters: (1) access to a computer and the Internet, with Zoom downloaded, and (2) residence in the city centre. Thereafter, the sampling methodology considered three main variables that were given equal weight in the selection process: country, city, and the extent to which the candidates' professional lives depended on tourism (see Figure A1). The sample size followed Saunders et al.'s [110] recommendations for heterogeneous purposive samples.

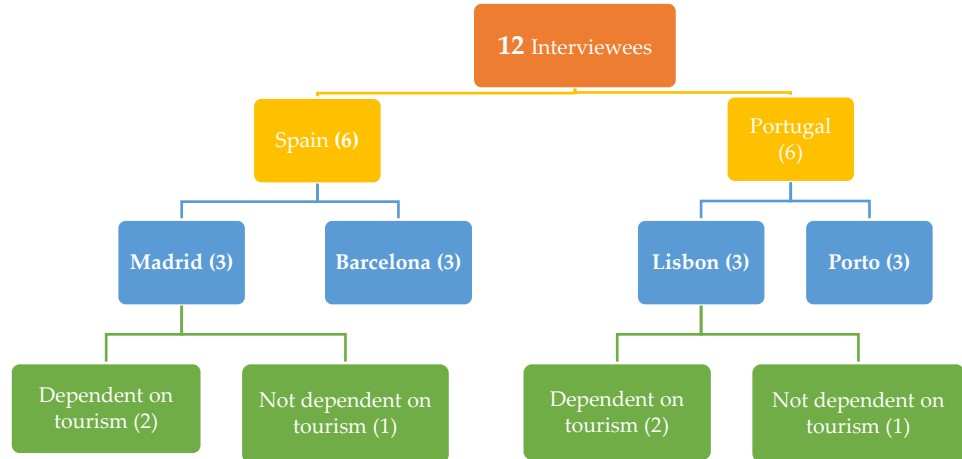

**Figure A1.** Sample structure and size.

**Table A1.** Sample profile.

| Interviewee | Dependent on Tourism | Age (Years) | Profession | Gender |
| --- | --- | --- | --- | --- |
| Madrid 1 | Yes | >30 | Hotel manager | Female |
| Madrid 2 | Yes | <30 | Hotel manager | Female |
| Madrid 3 | No | <30 | Biologist | Male |
| Barcelona 1 | Yes | >30 | Museum director | Male |
| Barcelona 2 | Yes | >30 | Restaurant manager | Male |
| Barcelona 3 | No | <30 | Communication director | Male |
| Lisbon 1 | No | >30 | Architect | Female |
| Lisbon 2 | Yes | >30 | Head of tourism division | Female |
| Lisbon 3 | No | >30 | Architect | Male |
| Porto 1 | Yes | >30 | Hotel manager | Female |
| Porto 2 | No | >30 | Senior social cohesion officer | Male |
| Porto 3 | Yes | <30 | Hotel manager | Female |

**Appendix B**

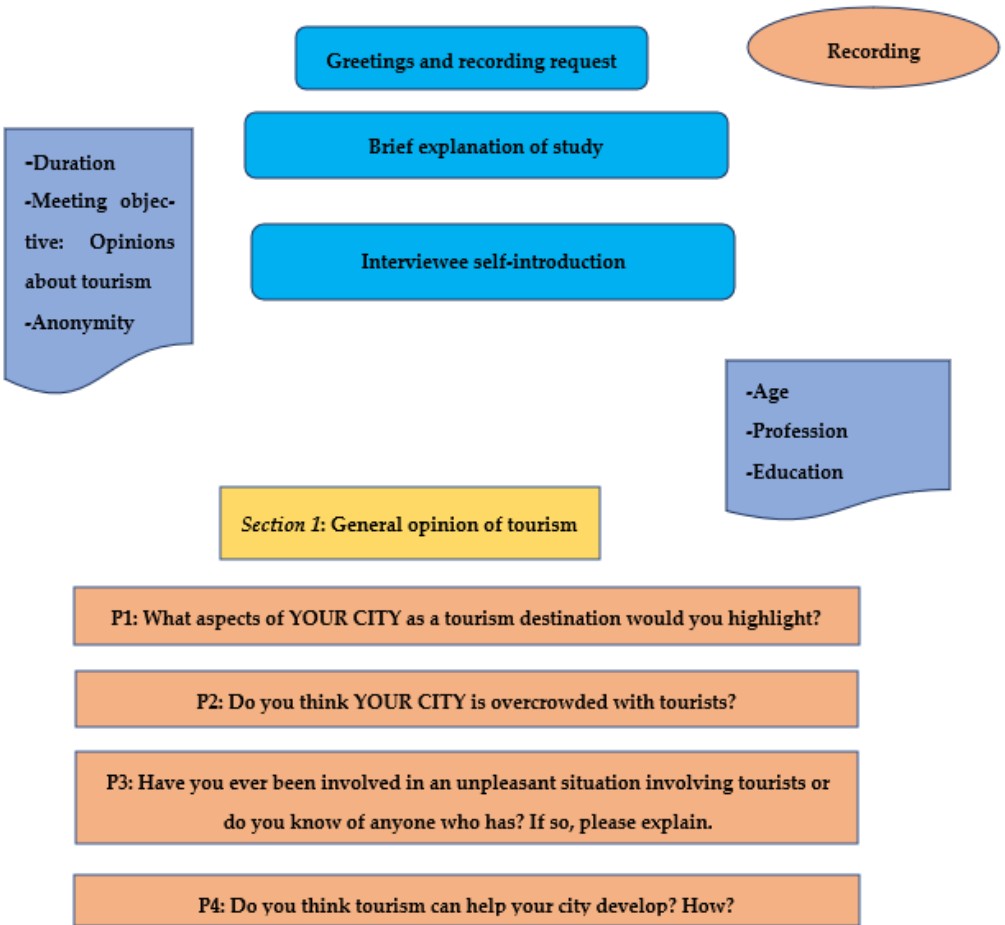

**Figure A2.** *Cont.*

**Section 2: Questions about five concepts**

P6: What do you think about smart cities? What do you think they are? (PAUSE) What characterises them or differentiates them from other cities?

P7: When you think of smart cities, which particular one(s) come(s) to mind? (PAUSE)

P8: What is well-being? (PAUSE)

P9: Now, which are the sources of well-being in YOUR CITY?

P8: What is well-being specifically for you? (PAUSE)

P9: Now, what are YOUR CITY's sources of well-being?

P10: And sources of ill-being?

**Section 3: COVID-19 and tourism**

P11: In your opinion, how has the COVID-19 pandemic affected tourism?

**Section 4: Future of tourism**

Projective technique 3: Construction technique

Now let's think about the future. What do you think sustainable cities of the future will be like? Please describe one to me.

P12: Finally, what, in your opinion, will tourism be like in your city 10 years from now?

**Figure A2.** Survey' Blocks.

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
