# Peer review of "Resilience and Sustainable Urban Tourism: Understanding Local Communities’ Perceptions after a Crisis"

_sustainability, doi:10.3390/su151813298_

Round 1
Reviewer 1 Report
Dear Authors,
An interesting manuscript that examines resilience and sustainable city tourism.
There are several observations that would improve the manuscript:
1. Instead of coronavirus disease-19, it is better to simply use COVID-19;
2. It is not clear what the purpose of the research is, although the questions are presented;
3. It is necessary to improve the clarification of the research gap, because now it is very bold to say that such research was lacking. For example, the following examples can be used for inspiration: (I) Dvorak, J. (2021). Response of the Lithuanian municipalities to the First Wave of COVID-19. Baltic Region, 13(1), 70-88; Kańduła, S., & Przybylska, J. (2021). Financial instruments used by Polish municipalities in response to the first wave of COVID-19. Public Organization Review, 21(4), 665-686.
4. It is not clear why the analysis of literature should begin with the introduction. Maybe 2.1 should start immediately after the title.
5. It is necessary to consider how to better describe sustainable tourism, because now little attention is paid to it, because tourism is usually not only a positive activity for the area, but can be the cause of garbage and noise. For inspiration, you can find an article about: Burksiene, V., Dvorak, J., & Burbulyte-Tsiskarishvili, G. (2018). Sustainability and sustainability marketing in competing for the title of European Capital of Culture. Organization, 51(1), 66-78.
6. The selection of cities based on population is not a convincing argument, at least some criteria are needed, because now, for example, it is not clear why Valencia was not chosen.
7. The authors argue that only 12 respondents can be selected. However, their cases are better called expert interviews, because most of the respondents work in the hospitality industry. However, it is not entirely clear why a biologist was chosen, architects, I don't really know what a senior social cohesion officer does. Because the research question is: What concepts or variables do residents perceive as being at the core of the COVID-19 pandemic's main impacts on city tourism?
8. Nothing is said about the ethics of qualitative interviewing.
9. 5.1 - 5.3 should be converted into conclusions
All the best
Author Response
Dear Authors,
An interesting manuscript that examines resilience and sustainable city tourism.
There are several observations that would improve the manuscript:
Thank you for your feedback. We have endeavoured to address all your concerns as described below.
Reviewer 1’s Comment 1
Instead of coronavirus disease-19, it is better to simply use COVID-19;
Authors’ response to Reviewer 1’s Comment 1
We appreciate your suggestion. However, we have elected to keep this definition in our manuscript to follow standard academic practice requiring that all abbreviations and acronyms be preceded by their full associated term when they first appear in a paper. This guideline ensures that all readers now and in the future can immediately understand the acronyms used.
Reviewer 1’s Comment 2
It is not clear what the purpose of the research is, although the questions are presented;
Authors’ response to Reviewer 1 Comment 2
We have added the following sentence to clarify our aims before presenting the research questions: ‘More specifically, the aim of this study is to understand the crisis’s perceived impacts on communities and residents directly or indirectly affected by city tourism.’
Reviewer 1’s Comment 3
- It is necessary to improve the clarification of the research gap, because now it is very bold to say that such research was lacking. For example, the following examples can be used for inspiration: (I) Dvorak, J. (2021). Response of the Lithuanian municipalities to the First Wave of COVID-19. Baltic Region, 13(1), 70-88; Kańduła, S., & Przybylska, J. (2021). Financial instruments used by Polish municipalities in response to the first wave of COVID-19. Public Organization Review, 21(4), 665-686.
Authors’ response to Reviewer 1’s Comment 3
We found these sources to be extremely useful as they reinforce the evidence for the research gap we sought to fill. The introduction now reads as follows:
Previous studies have assessed the impacts the financial intervention by municipalities to cope with COVID-19 (KaÅ„duÅ‚a and· Przybylska, 2021) and the types of economic intervention in different cities (Dvorak, 2021). This study offers a different perspective, by concentrating on residents’ perceptions of the impacts on their city’s tourism.
This study thus addressed a gap in the literature on the COVID-19 crisis’s effect on local communities’ resilience.
Reviewer 1’s Comment 4
It is not clear why the analysis of literature should begin with the introduction. Maybe 2.1 should start immediately after the title.
Authors’ response to Reviewer 1’s Comment 4
Per your suggestion, we have moved the 2.1. subheading to immediately after the 2. heading:
2.Literature Review
2.1. Sustainable Tourism Development and COVID-19 Pandemic’s Influence
Reviewer 1’s Comment 5
It is necessary to consider how to better describe sustainable tourism, because now little attention is paid to it, because tourism is usually not only a positive activity for the area, but can be the cause of garbage and noise. For inspiration, you can find an article about: Burksiene, V., Dvorak, J., & Burbulyte-Tsiskarishvili, G. (2018). Sustainability and sustainability marketing in competing for the title of European Capital of Culture. Organization, 51(1), 66-78.
Authors’ response to Reviewer 1’s Comment 5
We have rewritten our definition of sustainable tourism as follows:
Local communities must prioritise the development of resilience in the COVID-19 crisis’s wake and aim for more sustainable development and sustainable tourism in the future. Sustainable development requires an holistic approach that includes four pillars: social, economic, ecological and cultural (Burksiene & Burbulyte-Tsiskarishvili, 2018).
Reviewer 1’s Comment 6
The selection of cities based on population is not a convincing argument, at least some criteria are needed, because now, for example, it is not clear why Valencia was not chosen.
Authors’ response to Reviewer 1’s Comment 6
We have clarified that size was the criterion used to select the cities under analysis:
This study targeted the two largest cities of Portugal and Spain. Madrid and Barcelona were chosen as the Spanish cities for this study as they are the two most visited and populous Spanish cities. Madrid has 5,743,402 inhabitants and Barcelona 6,779,888 [77]. Their rivalry is well-known [79]. Following the same reasoning, the Portuguese cities selected were Lisbon (1,083,050 inhabitants) and Porto (837,555 inhabitants) [80]. The tourism industry in these four cities has suffered great losses due to the COVID-19 pandemic.
Reviewer 1’s Comment 7
- The authors argue that only 12 respondents can be selected. However, their cases are better called expert interviews, because most of the respondents work in the hospitality industry. However, it is not entirely clear why a biologist was chosen, architects, I don't really know what a senior social cohesion officer does. Because the research question is: What concepts or variables do residents perceive as being at the core of the COVID-19 pandemic's main impacts on city tourism?
Authors’ response to Reviewer 1 Comment 7
We have explained that the participants were recruited from different areas to cover COVID-19’s diverse positive and negative effects on local communities. Social cohesion officers are responsible for overseeing and managing programmes, initiatives or strategies focused on strengthening interrelationships within a community or society. Social cohesion in this research context thus refers to the degree to which individuals and groups are connected, included and willing to cooperate within communities.
Reviewer 1 Comment 8
- Nothing is said about the ethics of qualitative interviewing.
Authors ‘response to Reviewer 1 Comment 8
We have added a sentence about informed consent.
Reviewer 1 Comment 9
- 5.1 - 5.3 should be converted into conclusions
Authors ‘response to Reviewer 1 Comment 9
Thank you for this suggestion. We have added a new section containing all the conclusions.

Reviewer 2 Report
The article raises a very important issue, which is little described in the literature on the subject. Generally, other researchers focused on the impact of the pandemic on tourism during its duration.
However, it is worth conducting research that will explain what effects the pandemic had on local communities and local tourist industries after its completion.
Maybe during the restart of tourism, we will be able to avoid the problems that were the main problems before the outbreak of the pandemic.
The article meets all the requirements and is an important voice in the scope of knowledge analyzed by the authors.
Author Response
Reviewer 2’s Comments
The article raises a very important issue, which is little described in the literature on the subject. Generally, other researchers focused on the impact of the pandemic on tourism during its duration.
However, it is worth conducting research that will explain what effects the pandemic had on local communities and local tourist industries after its completion.
Maybe during the restart of tourism, we will be able to avoid the problems that were the main problems before the outbreak of the pandemic.
The article meets all the requirements and is an important voice in the scope of knowledge analyzed by the authors.
Authors’ reply to Reviewer 2’s Comments
Thank you for taking the time to read our paper. We greatly appreciate your positive feedback.

Reviewer 3 Report
Lines 34-35: you talk about suffering of local communities, but the named factors are only the economies of the local communities. In total, local communities have suffered the most due to illness and death in the pandemic- people loss and living quality loss due to health problems (thematized later in the article at lines 222-226). Also in the next sentence (lines 36 and 37) you jump to positive influence of a pandemic in terms of positivity, optimism and resilience. I really do not find it appropriate to connect positivity and optimism in local community with the pandemic in which numerous millions of people died and the pandemic-induced lockdowns that almost brought the economy to a halt.
Line 88: there must be some more current estimation, this is old data. See for example:
https://wttc.org/news-article/wttc-reveals-us-travel-tourism-sector-suffered-loss-of-766-billion-in-2020
Line 174: it is not only about the sustainability of tourism in itself but of the tourist destination as a context for the community to flourish through tourism. See for example:
Paunović, I., Dressler, M., Mamula Nikolić, T., & Popović Pantić, S. (2020). Developing a competitive and sustainable destination of the future: Clusters and predictors of successful national-level destination governance across destination life-cycle. Sustainability, 12(10), 4066.
Lines 634-637: For further information regarding regional brand image and community development priorities, see the article below. Also, an important aspect of community resilience through participation (lines 104-105) are the democratic nature of the participation processes in order to spur innovation and creativity for sustainable development and resilience.
Paunovic, I., Müller, C., & Deimel, K. (2023). Citizen Participation for Sustainability and Resilience: A Generational Cohort Perspective on Community Brand Identity Perceptions and Development Priorities in a Rural Community. Sustainability, 15(9), 7307.
Author Response
Reviewer 3’s Comment 1
Lines 34-35: you talk about suffering of local communities, but the named factors are only the economies of the local communities. In total, local communities have suffered the most due to illness and death in the pandemic- people loss and living quality loss due to health problems (thematized later in the article at lines 222-226). Also in the next sentence (lines 36 and 37) you jump to positive influence of a pandemic in terms of positivity, optimism and resilience. I really do not find it appropriate to connect positivity and optimism in local community with the pandemic in which numerous millions of people died and the pandemic-induced lockdowns that almost brought the economy to a halt.
Authors’ response to Reviewer 3’s Comment 1
We have rewritten this part of our paper to follow your insightful suggestion.
Reviewer 3’s Comment 2
Line 88: there must be some more current estimation, this is old data. See for example:
https://wttc.org/news-article/wttc-reveals-us-travel-tourism-sector-suffered-loss-of-766-billion-in-2020
Authors’ response to Reviewer 3’s Comment 2
Thank you for your comment. We have updated the data accordingly.
Reviewer 3’s Comment 3
Line 174: it is not only about the sustainability of tourism in itself but of the tourist destination as a context for the community to flourish through tourism. See for example: Paunović, I., Dressler, M., Mamula Nikolić, T., & Popović Pantić, S. (2020). Developing a competitive and sustainable destination of the future: Clusters and predictors of successful national-level destination governance across destination life-cycle. Sustainability, 12(10), 4066.
Authors’ response to Reviewer 3’s Comment 3
We have added the following sentence to address your concern: ‘Sustainability is of utmost importance to foster destination competitiveness and allows to improve socioeconomic conditions (Paunović et al., 2020).’
Reviewer 3’s Comment 4
Lines 634-637: For further information regarding regional brand image and community development priorities, see the article below. Also, an important aspect of community resilience through participation (lines 104-105) are the democratic nature of the participation processes in order to spur innovation and creativity for sustainable development and resilience.
Paunovic, I., Müller, C., & Deimel, K. (2023). Citizen Participation for Sustainability and Resilience: A Generational Cohort Perspective on Community Brand Identity Perceptions and Development Priorities in a Rural Community. Sustainability, 15(9), 7307.
Authors’ response to Reviewer 3’s Comment 4
We found the above research paper to be extremely useful, so we have added two new sentences to our article:
The study by Paunovic and Deimel (2023) claim for an intergenerational dialogue as regards perceived brand image attributes and development priorities to enhance citizen participation for sustainability and resilience.
According to previous research on tourism crises [30], community resilience positively affects locals’ crisis response and community participation. Community participation might be viewed as a democratic process doe the development of local policies that foreseen sustainability and resilience planning (Paunovic and Deimel, 2023).

Round 2
Reviewer 1 Report
Dear Authors,
You have implemented most of the corrections which where advised. I think the manuscript can be accepted to the Sustainability journal.
All the best
Author Response
Review 1’s Comments
Dear Authors,
You have implemented most of the corrections which where advised. I think the manuscript can be accepted to the Sustainability journal.
All the best
Authors’ reply to Reviewer 1’s Comments
Thank you for taking the time to read our paper. We greatly appreciate your positive feedback.
Reviewer 3 Report
Dear authors,
thank you for providing the revised version of the article. The article is significantly improved, but still needs a modest revision in order to better explain the methodology as well as discussion and conclusions.
The qualitative methodology needs to be better explained in terms of grounded research or similar approach. See for example:
Dunne, C. (2011). The place of the literature review in grounded theory research. International journal of social research methodology, 14(2), 111-124.
Mohajan, D., & Mohajan, H. (2023). Glaserian Grounded Theory and Straussian Grounded Theory: Two Standard Qualitative Research Approaches in Social Science.
Conclusions and discussion need to be rewritten. The conclusions section should contain less citations and more author’s original insights, thourghts and opinions based on the presented research (some paraphrased text parts can be moved to discussion section).
The discussion section needs to be better structured in terms of clearly differentiating between implications for research (the answer to the research questions 1 and 2), implications for practice and future research directions. The implications for practice of city destination management needs to be better presented.
Good luck with the changes!
Author Response
thank you for providing the revised version of the article. The article is significantly improved, but still needs a modest revision in order to better explain the methodology as well as discussion and conclusions.
Authors’ response to Reviewer 3’s Comment 1
Thank you for your positive feedback. We have endeavoured to address all your concerns as described below.
Reviewer 3’s Comment 2
The qualitative methodology needs to be better explained in terms of grounded research or similar approach. See for example:
Dunne, C. (2011). The place of the literature review in grounded theory research. International journal of social research methodology, 14(2), 111-124.
Mohajan, D., & Mohajan, H. (2023). Glaserian Grounded Theory and Straussian Grounded Theory: Two Standard Qualitative Research Approaches in Social Science. DOI: 10.1080/13645579.2010.494930
Authors’ response to Reviewer 3’s Comment 2
We have clarified that Leximancer software expediated the inductive identification of significant themes and that a major advantage of this approach was greater consistency in terms of applying grounded theory. We have also added the suggested references. The rewritten text reads as follows:
More specifically, Leximancer facilitated an inductive identification of themes, with minimal manual intervention, based on a two-step approach: conceptual analysis (i.e. frequency of concepts) and relational analysis (i.e. co-occurrence between concepts). The lexical-text analysis performed by Leximancer has also proven to offer consistent results (Harwood et al., 2015) with the grounded theory methodology (Dunne, 2011; Mohajan & Mohajan, 2023).
Reviewer 3’s Comment 3
Conclusions and discussion need to be rewritten. The conclusions section should contain less citations and more author’s original insights, thourghts and opinions based on the presented research (some paraphrased text parts can be moved to discussion section).
The discussion section needs to be better structured in terms of clearly differentiating between implications for research (the answer to the research questions 1 and 2), implications for practice and future research directions. The implications for practice of city destination management needs to be better presented.
Authors’ response to Reviewer 3’s Comment 3
We have rewritten the discussion and conclusions as requested. The former section now includes the answer to each research question and a discussion of how our results compare with previous research’s findings. The conclusions have been divided into three subsections: theoretical contributions, practical implications, and limitations and avenues for future research.